# Predicting malaria outbreak in The Gambia using machine learning techniques

**Ousman Khan[1], Jimoh Olawale Ajadi[1,2]\*, M. Pear Hossain[3,4]**

**1** Department of Mathematics, College of Computing and Mathematics, King Fahd University of Petroleum and Minerals, Dhahran, Saudi Arabia, **2** Interdisciplinary Research Center for Refining & Advanced Chemicals, King Fahd University of Petroleum and Minerals, Dhahran, Saudi Arabia, **3** WHO Collaborating Centre for Infectious Disease Epidemiology and Control, School of Public Health, Li Ka Shing Faculty of Medicine, The University of Hong Kong, Hong Kong Special Administrative Region, China, **4** Laboratory of Data Discovery for Health Limited (D24H), Hong Kong Science and Technology Park, New Territories, Hong Kong Special Administrative Region, China

These authors contributed equally to this work.

\* jimoh.ajadi@kfupm.edu.sa

**Data Availability Statement:** The clinical data is available from Health Management Information System (HMIS) office under the ministry of health in The Gambia. The program's manager can be reached on this email: Abdoulie52000@yahoo.

## Abstract

Malaria is the most common cause of death among the parasitic diseases. Malaria continues to pose a growing threat to the public health and economic growth of nations in the tropical and subtropical parts of the world. This study aims to address this challenge by developing a predictive model for malaria outbreaks in each district of The Gambia, leveraging historical meteorological data. To achieve this objective, we employ and compare the performance of eight machine learning algorithms, including C5.0 decision trees, artificial neural networks, k-nearest neighbors, support vector machines with linear and radial kernels, logistic regression, extreme gradient boosting, and random forests. The models are evaluated using 10-fold cross-validation during the training phase, repeated five times to ensure robust validation. Our findings reveal that extreme gradient boosting and decision trees exhibit the highest prediction accuracy on the testing set, achieving 93.3% accuracy, followed closely by random forests with 91.5% accuracy. In contrast, the support vector machine with a linear kernel performs less favorably, showing a prediction accuracy of 84.8% and underperforming in specificity analysis. Notably, the integration of both climatic and non-climatic features proves to be a crucial factor in accurately predicting malaria outbreaks in The Gambia.

## Introduction

Malaria, a parasitic disease transmitted primarily by mosquitoes, continues to pose a significant threat to public health, particularly in tropical and subtropical regions [1]. It manifests with symptoms ranging from minor to severe, often leading to fatal outcomes, with an estimated 627,000 lives lost in 2020 alone, accounting for 7.8% of childhood mortality globally [2]. Despite substantial investments in malaria control and research programs by the World Health Organization (WHO) in recent years [3], malaria remains a prominent concern in Sub-Saharan Africa, notably affecting children under the age of five [4].

com. The climatic data was obtained from the Department of Water Resources, the meteorological division under the Ministry of Fisheries, Water Resources and National Assembly Matters in The Gambia. The office can be contacted using this email: info@mofwr.gov.gm.

**Funding:** This work was supported by the Deanship of Research Oversight and Coordination at King Fahd University of Petroleum and Minerals. The funders had no role in study design, data collection and analysis, decision to publish, or preparation of the manuscript.

**Competing interests:** The authors have declared that no competing interests exist.

In The Gambia, malaria prevalence has been associated with specific climatic conditions favorable to the growth of the Plasmodium parasite and the Anopheles mosquito [5]. Although malaria cases have decreased over the years, the disease's highly seasonal transmission persists, primarily during and immediately after the rainy season [6]. The Gambia National Malaria Control Program (GNMCP) has played a pivotal role in reducing the country's malaria burden, leading to a decrease in the number of cases from 346.9 per 100,000 people in 2004 to 66 per 100,000 people in 2018 [7]. However, challenges persist, particularly in the eastern region of The Gambia, where malaria prevalence remains high, ranging from 10 to 40% [8]. This imposes a substantial financial and health burden on households in this region [9]. The persistence of intense seasonal transmission during and after the rainy season underscores the need for effective preventative measures.

Machine learning's transformative impact on healthcare is evident in its ability to swiftly analyze datasets and enhance decision-making for improved patient outcomes [10]. While traditionally associated with large datasets, recent applications, such as a study on juvenile-onset systemic lupus erythematosus [11], demonstrate machine learning's effectiveness in extracting insights from smaller datasets. This adaptability positions machine learning as a versatile tool in healthcare, influencing disease prognosis, prediction, and other critical tasks.

Forecasting malaria outbreaks using machine learning poses several challenges. Despite the promising capabilities of machine learning techniques in healthcare, accurately predicting the occurrence and dynamics of malaria outbreaks is intricate due to the complex interplay of various factors. Climatic conditions, socio-economic factors, and human behavior contribute to the complexity of the malaria transmission dynamics, making it a multifaceted challenge for predictive modeling. This study addresses these challenges by leveraging machine learning algorithms to gain valuable insights into the impact of climatic factors on malaria outbreaks in The Gambia. The utilization of C5.0 decision trees (DT), artificial neural networks (ANN), k-nearest neighbors (KNN), support vector machines (SVM) with linear (svmLinear) and radial kernels (svmRadial), logistic regression (LR), extreme gradient boosting (XGBOOST), and random forests (RF) reflects an effort to overcome these complexities and contribute to the advancement of malaria control and early warning systems.

## Related work

Various studies have used and compared different machine learning approaches to predict malaria outbreak and total malaria cases using meteorological data and malaria cases. Studies have revealed that climate has a significant impact on the incidence and spread of malaria, however the impact of different climate variables varied by region [5]. The two main factors that contributes to the transmission of malaria are rainfall and temperature [12]. Additionally, temperature, rainfall, and relative humidity are the main climatic variables that influence the plasmodium parasite's growth and maturity, therefore any change in these variables would undoubtedly have an impact on mosquito ecology [13].

Kalipe et al. [14] explored the prediction of malaria outbreaks in healthcare centers in the Visakhapatnam district, India. They conducted a comprehensive analysis using historical meteorological data and records of malaria cases collected over six years. Specifically, the study involved the combination and aggregation of two distinct datasets: one comprising historical meteorological data and the other consisting of records of malaria cases. Among the various machine learning models considered, XGBoost stood out, demonstrating superior performance with an accuracy of 96.26%. This finding emphasizes the effectiveness of XGBoost in comparison to the other models assessed in the study.

Zinszer et al. [15] predicts malaria using environmental and clinical indicators across different settings in Uganda. Multivariate autoregressive integrated moving average models were used to create facility-specific forecasting models for confirmed malaria, which generated weekly forecast horizons across a 52-week forecasting period. They came to the conclusion that, when combined with environmental indicators, clinical data could be utilized to increase the accuracy of malaria forecasts in endemic environments. Similarly, Lee et al. [16] conducted a study using clinical data extracted from PubMed abstracts spanning from 1956 to 2019. The focus of their study was to compare the predictive performances of six machine learning models: SVM, RF, Multilayer Perceptron, AdaBoost, Gradient Boosting (GB), and CatBoost for malaria prediction. Addressing data imbalance concerns, they applied the Synthetic Minority Oversampling Technique (SMOTE). Post-SMOTE implementation, Random Forest emerged as the most effective model. The outcomes of their investigation led to the conclusion that leveraging patient data with machine learning techniques enables accurate malaria prediction.

Stephen et al. [17] used 5 supervised machine learning techniques, including naive Bayes (NB), support vector machine, linear regression, logistic regression (LR), and KNN, to model the outbreak of malaria. Results showed that NB had the best accuracy for both training and testing, with an average accuracy of 79.1%. Adamu et al. [18] compared six different machine learning classifiers for the prediction of malaria cases using weather, non-climatic characteristics, and malaria cases: SVM, KNN, RF, DT, LR, and NB. With an accuracy of 97.72% on the data set utilized for the investigation, it was discovered that RF is the best model. The findings indicated that non-climatic and atmospheric factors play a substantial role in predicting the frequency of malaria outbreaks in a certain society.

In [19], the authors aimed to develop a hybrid classification and regression model for predicting disease outbreaks using datasets, focusing on malaria outbreaks. The motivation behind this approach stemmed from the recognition that certain single data mining techniques faced accuracy challenges. The hybrid model, created by combining decision trees, random forests, Naive Bayes multinomials, simple logistic, Sequential Minimal Optimization (SMO), instant-Base learning algorithm (IB1), and Bayesian logistic regression, demonstrated improved output results compared to employing a single technique. During hybrid training, the model achieved an accuracy of 100%, and during hybrid evaluation, it maintained an accuracy of 75%.

In Senegal, Diao et al. [20] conducted a significant study in malaria forecasting. They formulated a generalized linear model based on Poisson and negative binomial regression models, considering climatic variables, insecticide-treated bed-nets distribution, Artemisinin-based combination therapy, and historical malaria incidence. The study demonstrated the efficacy of the Poisson regression model and addressed issues of over-forecasting through the saturation of rainfall. In Rajasthan, India, Singh et al. [21] proposed a hybrid ML algorithm (Probabilistic Principal Component Analysis $P^2$CA – Particle Swarm Optimization (PSO) – ANN) for malaria outbreak prediction in districts like Barmer, Bikaner, and Jodhpur. Using meteorological variables, data fusion, and $P^2$CA, the model achieved accurate predictions, outperforming benchmarks. It shows promise as an early warning system based solely on meteorological data.

Despite these valuable contributions, several limitations persist in the existing research. Challenges related to data imbalance, model generalization, and regional variations in climatic impacts on malaria transmission are frequently encountered. Many studies face difficulties in achieving a balance in the distribution of malaria cases, affecting the performance of predictive models. Additionally, the generalization of models to diverse geographic regions remains a challenge, as the impact of climatic variables on malaria transmission may vary significantly. Furthermore, inconsistencies in methodologies and datasets hinder direct comparisons across studies.

It is essential to note these limitations, as they underscore the need for a nuanced and context-specific approach in the development of malaria prediction models. More importantly, the identification of these challenges emphasizes the significance of the present study in addressing and contributing to the resolution of these limitations in the literature. In addition, to the best of our knowledge, this is the first study on the use of machine learning techniques to predict malaria outbreaks in The Gambia.

## Materials and methods

In conducting the analysis for prediction, we utilized a robust set of tools and software to ensure accuracy and reliability. All the statistical procedures conducted in this study were implemented using the most recent version of R, a widely recognized open-source statistical tool known for its versatility and extensive data analysis capabilities. Specifically, the predictive modeling and analysis were performed using the caret package [22] in R. The package provides a comprehensive framework for building, training, and evaluating predictive models, making it particularly suitable for our study's objectives.

### Study area

The Gambia is the smallest mainland nation in Africa with an estimated population of 2,603,917 people. It has a total land area of 11,300 $km^2$ and is situated on the west coast of Africa with latitude and longitude of $13°28'N$ and $16°34'W$ respectively. The country is bordered by Senegal on three sides and with a narrow Atlantic coastline on the west.

The country is divided into 8 local government areas and subdivided into 43 districts. And is home to at least ten distinct ethnic groups, each of which has maintained its own language and extensive cultural heritage.

The climate in The Gambia is characterized by a prolonged dry season lasting from mid-October to mid-June, succeeded by a brief rainy season from June to September, with an average daily temperature of $28°C$ conducive to the survival of Anopheles gambiae mosquitoes [6]. The average temperature ranges from 18 to $30°C$ during the dry season and from 23 to $33°C$ during the wet season. Annual rainfall averages from 700 millimeters in the far north to 1, 000 millimeters in the South and Southeast.

Malaria exhibits a seasonal pattern in The Gambia, with the highest incidence of clinical cases and fatalities occurring between September and November [23]. Additionally, the majority of malaria cases in The Gambia are reported during the rainy season and the subsequent one or two months.

### Dataset used

The specific contribution of our study lies in utilizing two distinct datasets (historical meteorological and clinical datasets) spanning nine years (January 2013 to December 2021) to predict malaria outbreaks in each district of The Gambia. While previous studies have highlighted the influence of climatic factors such as rainfall, temperature, and humidity on malaria incidence [12, 24], our approach integrates machine learning techniques to provide a more accurate and district-specific prediction of malaria outbreaks. This extends the existing knowledge by leveraging advanced analytical methods for a targeted and nuanced understanding of the impact of climatic conditions on malaria transmission at a local level.

The clinical dataset obtained from the health management information system (HMIS), Directorate of Planning and Information under the ministry of health contains monthly records of malaria cases, both severe and uncomplicated cases and total deaths for each district. These data were collected from all district hospitals and clinics across the country by well-

## A map of the Gambia showing major meteorological stations

**Fig 1. The map of the Gambia showing its five main meteorological stations across different regions.**

trained nurses (state registered or state enrolled) and doctors. The majority of instances of malaria were uncomplicated. The cases and deaths were recorded per month for each district from January 2013 to December 2021. The population size for each district was also incorporated in this dataset. The total population for 2013 was taken from the 2013 population and housing census conducted by The Gambia Bureau of Statistics (GBOS), while the remaining years were projected total for each district.

The meteorological data were collected from the Department of water resources, the meteorological division under the Ministry of Fisheries, water resources and national assembly matters. The data contains three main climatic variables; temperature, rainfall and relative humidity. The minimum and maximum temperatures were measured in degree Celsius (˚C), relative humidity in percent (%) and rainfall in millimeters (*mm*). The monthly average readings were taken from 9 weather stations spread across the country. Fig 1, created using spData package [25] and ggspatial package [26] in R software, depicts the locations of the main five out of these nine weather stations in The Gambia, situated across different regions of the country.

### Data cleaning and preprocessing

After combining the clinical and meteorological datasets, the final dataset contains 4428 observations and nine variables. Rigorous steps were taken to ensure the dataset's integrity and reliability. Fortunately, there were no missing values for crucial variables, such as, deaths, severe cases, uncomplicated cases, and population size. However, approximately 4% of the meteorological data, including minimum temperature, maximum temperature, rainfall, and relative humidity, had missing observations. To address these missing values, we employed the mean imputation technique, calculating the mean of non-missing values for the respective month across the years 2013 to 2021. This meticulous imputation approach aimed to maintain data quality and consistency.

Moreover, the dataset underwent a comprehensive normalization process to enhance the effectiveness of machine learning algorithms. It is a critical preprocessing step that plays a pivotal role in enhancing the quality and effectiveness of machine learning algorithms. In this study, normalization was applied to numerical independent variables using the min-max normalization technique. This process transforms features into a shared range, mitigating the

impact of larger numeric values dominating the model's learning process.

$$\frac{X - X_{min}}{X_{max} - X_{min}} \tag{1}$$

Where $X_{max}$ and $X_{min}$ represent the variables' maximum and minimum values, respectively. After scaling, all the numerical variables have values between 0 and 1.

Normalization becomes particularly crucial in scenarios where the scale of numerical features varies significantly. Without normalization, features with larger numeric values may exert undue influence on the learning algorithm, potentially overshadowing the contributions of smaller-scale features. The rationale behind data normalization lies in its ability to create a level playing field for numerical features, ensuring that each contributes proportionally to the learning process. This contributes to the overall robustness and reliability of the machine learning models employed in predicting malaria outbreaks. The normalization formula employed in this study, as depicted in Eq 1, ensures that all numerical variables are scaled to values between 0 and 1.

Several researchers have emphasized the importance of data normalization in improving data quality and overall algorithm performance. By implementing this preprocessing technique, biases associated with feature scales are minimized, allowing machine learning models to more effectively discern patterns and relationships within the data [27].

In addition to addressing missing values and normalizing features, the dataset was carefully examined for potential anomalies and outliers. The final step involved creating the dependent variable, named "outbreak," based on a carefully designed ratio using malaria cases and population size as shown in Eq 2.

$$r = \frac{C_s + C_u}{P} \tag{2}$$

where, $C_s$ and $C_u$ represent the severe and uncomplicated cases, respectively and $P$ denotes the population size in each district in the Gambia. To categorize the ratio into meaningful groups, an outbreak was defined as a ratio of cases greater than 0.01, signifying a significant threat in this context. Conversely, a no outbreak was defined as a ratio of cases less than or equal to 0.01.

The combination of these steps not only addressed missing data but also ensured the overall reliability and quality of the dataset. This thorough preprocessing lays a strong foundation for the subsequent machine learning analyses.

Fig 2 shows the distribution of all the numeric variables in the dataset. Clearly, malaria cases, deaths, and total rainfall are highly skewed to the right. While maximum temperature appear to be slightly normally distributed, minimum temperature appeared to be left skewed.

## Prediction models

We applied eight different algorithms; ANN, KNN, C5.0 DT, support vector machine using linear kernel (svmLinear), support vector machine using radial kernel (svmRadial), XGBoost, RF and LR to predict malaria outbreak in each district in the Gambia. Chosen for their widespread use and high predictive accuracy, these models are versatile and commonly employed in both classification and regression tasks. In the context of machine learning, classification involves categorizing instances into predefined classes, such as predicting whether a district is likely to experience a malaria outbreak. On the other hand, regression predicts continuous numerical values. Our study primarily focuses on a classification problem.

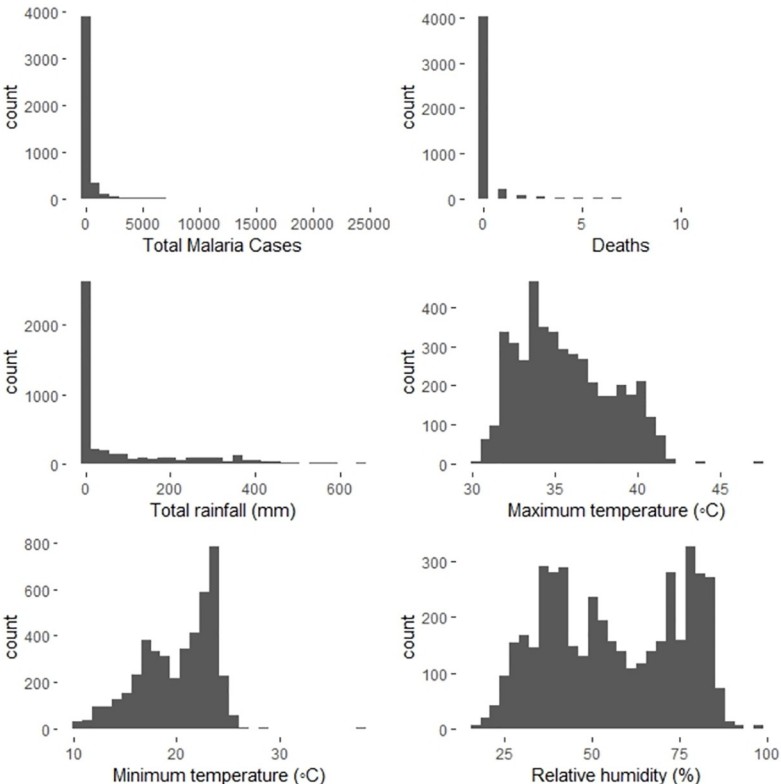

**Fig 2. Distribution of climatic variables, malaria cases and deaths: Malaria cases, deaths and rainfall are highly skewed to the right.** None of the features appeared to be distributed normally.

Initially, the dataset was divided into a training set (70%) and a testing set (30%). The features were extracted from the training set during the model training phase. Subsequently, the testing set was utilized to evaluate how each model performed in predicting malaria outbreaks.

- **ANN:** The Artificial Neural Network (ANN) is inspired by the functioning of the biological nervous system. It comprises interconnected nodes arranged in three layers: input, hidden, and output. Neurons in the input layer distribute the input signal to neurons in the hidden layer, and the output is determined similarly. The basic structure involves synthetic neurons and a mathematical function [28–30]. Let $x_i$ represent the input, $w_{ij}$ the weights, and $f$ the activation function, then the output $y_k$ can be calculated as:

$$y_k = f\left(\sum_i w_{ij}x_i\right)$$

- **KNN:** The KNN algorithm is one of the earliest and most straightforward algorithms [31] which can be applied in both classification and regression tasks. KNN classifies data points based on the nearest training data points. It is non-parametric and classifies based on proximity without considering dataset dimensions [31, 32]. Let $d$ represent the distance metric, $k$ the number of neighbors, and $c_i$ the class, then the prediction $\hat{y}$ for a new data point can be

expressed as:

$$\hat{y} = \arg \max_{c_i} \sum_j I(d(x_j, x_{new}) < k)$$

- **DT:** Decision Trees (DT) are hierarchical structures with root, branches, internal nodes, and leaf nodes. Using a divide and conquer method, the ideal split points are found using a top-down greedy search [33–35]. Let $S$ be the set of samples, $X$ the set of features, and $H(X)$ the information content, then the DT algorithm aims to recursively partition $S$ into subsets $S_i$ using $X$ and $H(X)$.
  The most well-known decision tree algorithms are Breiman et al.'s CART [35] and Quinlan's ID3, C4.5, and C5.0 [36, 37]. However, we used the C5.0 algorithm in this study.

- **RF:** Random Forest (RF) is an ensemble technique relying on multiple decision trees [38]. When DTs are grown very deeply, the training data are frequently overfitted, leading to a significant fluctuation in classification results for a little change in the input data. They become extremely sensitive to the training data as well as error-prone to the dataset as a result. RF solve this issue by training different parts of the training dataset using multiple DTs, and this helps in reducing bias, tolerate outliers, avoid overfitting and it is much less sensitive to the training data [39]. Let $T$ be the set of decision trees, $p_i$ the prediction of each tree, and $\hat{y}$ the final prediction, then the RF prediction can be formulated as:

$$\hat{y} = \arg \max_{c_i} \sum_j I(p_{ij} = c_i)$$

- **SVM:** Support Vector Machines (SVM) are effective for classification and regression. They have been used successfully in a variety of fields, including the medical field [39, 40]. Let $\mathbf{X}$ be the input data, $\mathbf{w}$ the weights, $b$ the bias, and $\epsilon$ the margin, then the SVM objective function for a linear kernel is given by:

$$\min_{\mathbf{w},b} \frac{1}{2} \|\mathbf{w}\|^2 + C \sum_i \max(0, 1 - y_i(\mathbf{w} \cdot \mathbf{X}_i - b))$$

  In this study, we utilized both the SVM model using a linear kernel (svmLinear) and SVM using radial kernel (svmRadial).

- **XGBoost:** XGBoost is an efficient ensemble technique based on gradient boosting. It minimizes a cost objective function, $J$, comprising a regularization term and a loss function [41]. The update rule for each iteration can be expressed as:

$$\hat{y}_{k+1} = \hat{y}_k - \eta \frac{\partial J}{\partial \hat{y}_k}$$

- **LR:** Logistic Regression (LR) models binary variables, which typically indicate whether an event will occur or not [42]. Let $\mathbf{X}$ be the input data, $\mathbf{w}$ the weights, $b$ the bias, and $\sigma$ the

sigmoid function, then the LR prediction can be written as:

$$\hat{y} = \sigma(\mathbf{w} \cdot \mathbf{X} + b)$$

### Feature selection

Feature selection is an important step in building and using machine learning models especially in disease prediction. It can greatly improve the performance of the model [43]. We use the backward elimination method in each model to select only the relevant climatic features in predicting malaria outbreak. This method helps in reducing overfitting, as well as save a significant amount of time during the learning process [44].

### Performance measures

We assessed each classification model's performance using a variety of performance metrics, including sensitivity, specificity, accuracy, and area under the receiver operating characteristic curve (ROC). The confusion matrix with parameters true positive (tp), true negative (tn), false positive (fp), and false negative (fn) is used to construct the performance measures of all the classifiers. The performance measures (accuracy, sensitivity and specificity) were calculated using these parameters. Below, we give an introduction and definition of each assessment criteria used to assess each learning model included in this study.

- **Accuracy**: It is the most widely used metrics for evaluating the performance of classifiers. It is the proportion of subjects with accurate labels to the entire group of subjects.

$$Accuracy = \frac{tp + tn}{tp + tn + fp + fn}$$

Accuracy is a widely adopted metric for classifier performance assessment.

- **Sensitivity**: The ability of a model to correctly identify positive examples in a binary classification is measured by sensitivity. It is also known as the recall or **true positive rate (TPR)**.

$$Sensitivity = \frac{tp}{tp + fn}$$

Sensitivity emphasizes the accurate identification of positive cases, contributing to a model's robustness. A model performs better when the sensitivity is higher at accurately identifying positive cases.

- **Specificity**: Specificity is the proportion of true negatives the model successfully detects. This percentage is also known as the **true negative rate (TNR)**.

$$Specificity = \frac{tn}{tn + fp}$$

Specificity complements sensitivity by highlighting the model's effectiveness in identifying true negatives.

- **Area under the ROC curve (AUC)**: The ROC curve is a plot that compares true positive rate (TPR) and false positive rate (FPR) at various classification thresholds. TPR is *sensitivity*

while FPR is 1 – *specificity*. AUC is a discrimination metric indicating how well the predictor can distinguish subjects into two categories. AUC provides a comprehensive overview of the model's discriminatory power across different threshold levels, offering insights into overall predictive performance.

These selected performance metrics collectively offer a thorough evaluation of the models, capturing aspects of accuracy, sensitivity, specificity, and discriminatory capability. Their inclusion ensures a multifaceted assessment aligned with the diverse requirements of malaria outbreak prediction.

## Upsampling

In the context of training and testing data, the initial random split of the dataset into these sets highlighted a significant class imbalance, particularly regarding "outbreak" and "no outbreak" cases, as depicted in Fig 3. Notably, 87% of both the training and testing datasets consisted of "no outbreak" cases.

To address the challenge posed by class imbalance during the training phase, we employed the up-sampling method exclusively on the training dataset. Up-sampling involves the generation of additional instances of the minority class (outbreak) through random sampling with replacement. The goal is to augment the size of the minority class to align it with the majority class (no outbreak) in the training data. Importantly, this up-sampling process is specifically applied to the training dataset and not the testing dataset.

By implementing up-sampling in this manner, the training data becomes more representative of the underlying distribution of outbreak and no outbreak cases. This contributes to a more balanced learning process during model training, allowing the machine learning algorithm to better capture patterns within the minority class. It's essential to note that up-sampling is a training-specific technique and does not directly influence the distribution of classes in the testing dataset or during cross-validation.

In summary, the use of up-sampling in the training dataset helps mitigate the challenges posed by class imbalance, fostering improved model generalization and predictive accuracy during the training phase while preserving the natural distribution of classes in the testing dataset and during cross-validation.

## Model validation

To robustly assess the performance of each model during the training phase, we employed a 10-fold cross-validation. This involved randomly dividing the training data into ten mutually exclusive folds of roughly equal size. Each model was then trained and tested ten times, with each iteration using nine subsets for training and the remaining subset for testing. This process yielded ten performance estimates, such as model accuracy, for each algorithm. The final performance estimate for each model was based on the average of these ten estimates, as shown in Eq 3.

$$CV_{PE} = \sum_{i=1}^{10} PE_i \tag{3}$$

where $PE_i$ stands for the performance estimate of fold $i$ and $CV_{PE}$ stands for the cross-validation performance estimate.

The choice of 10-fold cross-validation was motivated by its effectiveness in providing a robust evaluation of model performance. By dividing the training data into ten folds, our models underwent comprehensive training and testing iterations, promoting thorough exposure to

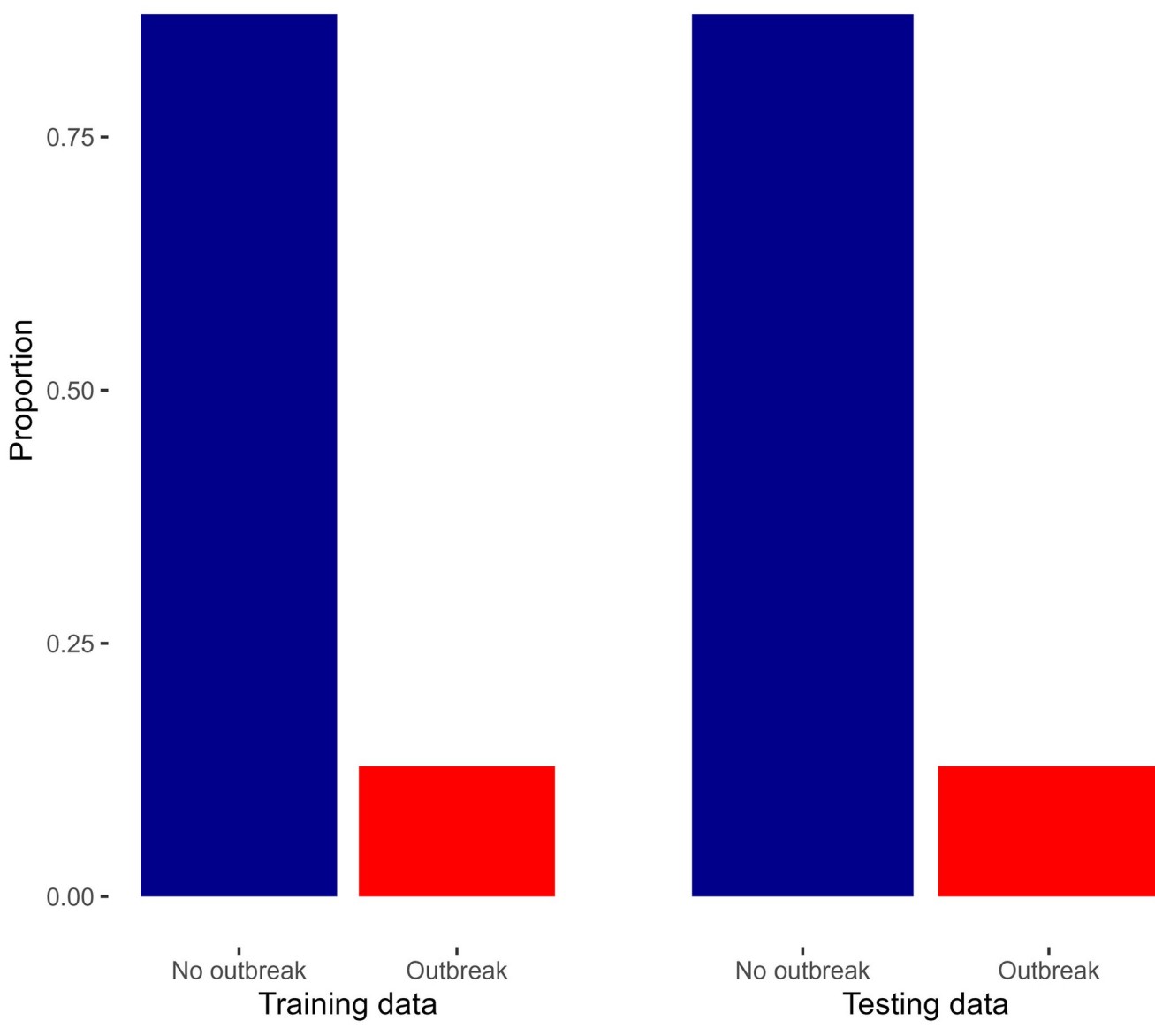

**Fig 3. Proportion of outbreak in both the training and testing data.**

diverse data patterns. This approach mitigates overfitting, enhances the reliability of performance estimates, and ensures a resource-efficient use of the available data. The decision to repeat the 10-fold cross-validation five times further strengthens the statistical significance of our performance evaluations.

**Table 1. Summary of all the variables in the dataset.**

| Categorical variable name | Distinct classes | | |
|---|---|---|---|
| Outbreak | 2 | | |
| Month | 12 | | |
| District | 41 | | |
| **Continuous variable name** | **Mean** | **S.D** | **Range** |
| Total rainfall | 77.80595 | 129.1016 | 0–651.4 |
| Minimum temperature | 20.13897 | 3.585981 | 10–37.5 |
| Maximum temperature | 35.64957 | 2.814196 | 30.2–47.36667 |
| Relative humidity | 56.42773 | 19.0511 | 17–98 |
| Population size | 53074.01 | 88917.42 | 3789–528789 |

## Results

### Summary of the dataset

The dataset utilized in this study comprises a diverse set of variables aimed at comprehensively understanding the factors influencing the occurrence of outbreaks. These variables encompass meteorological features, month, district and population size for each district in The Gambia. Table 1 presents a descriptive compilation of the variables used in this study.

### Climatic variables and malaria incidence patterns

The Pearson correlation among the climatic variables is shown in Fig 4. Most of the variables are either weakly or moderately correlated except between rainfall and relative humidity with a correlation of 0.7. The correlation between maximum temperature and relative humidity is also slightly strong with a correlation of −0.66. More importantly, all the results were found to be statistically significant using a significant level of 5% as shown in Table 2. These nuanced findings emphasize the complexity of the interactions between climate and malaria outcomes in the country.

Most of the clinical malaria cases occurred during the ending of the rainy season and the two months after, which is around August to December as shown in Fig 5. This is the time of year when there is a lot of greenery and stagnant water in the bushes and on the streets, which is ideal for anopheles mosquito reproduction. This goes on to further explain rainfall's role in forecasting malaria outbreaks. Except for December, these months also have the highest average relative humidity. Fig 5 further shows that over these months, the minimum and maximum temperatures ranged from 15.8 to $22.9°C$ and 32.25 to $35.33°C$, respectively.

### Model evaluation and performance comparison

We selected features for each model using backward elimination. Features included in the final models are presented in Table 3. The variables month, district and population size were set to be included in every model whereas the climatic variables were selected using the backward elimination method. Notably, both XGBoost and ANN were the only models that incorporated all climatic predictors. In the case of C5.0 DT and svmLinear, only one climatic variable was omitted. Conversely, svmRadial did not include any climate predictor.

To optimize the performance of each predictive model, hyperparameter tuning was conducted using the grid search method. This approach involves systematically searching through a predefined set of hyperparameter values to identify the combination that yields the optimal model performance.

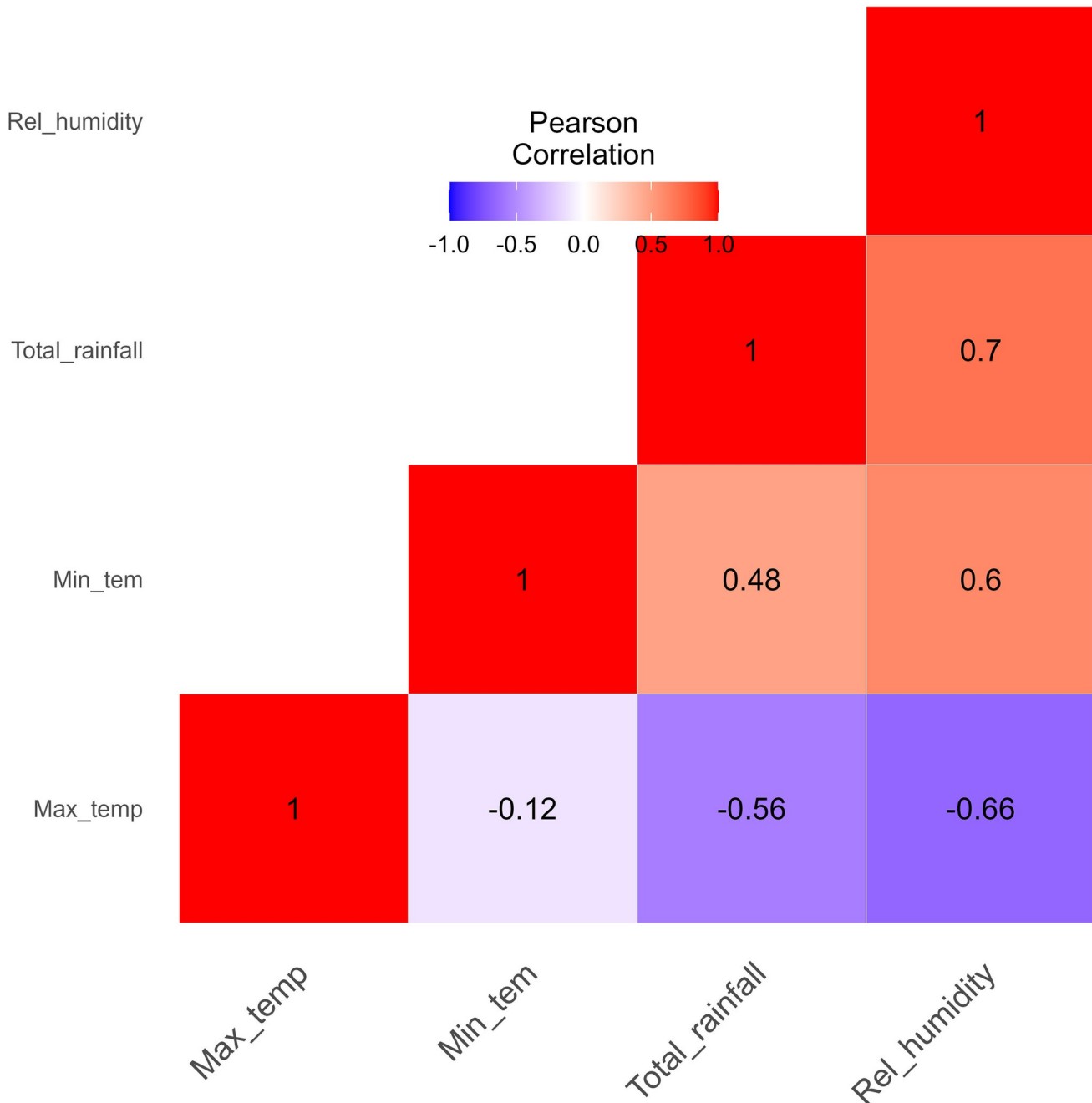

**Fig 4. The Pearson's correlation among all the climatic variables used.**

For instance, the parameter $k$ in the K-Nearest Neighbors (KNN) classifier was explored across the values $K = \{3, 5, 7, 9, 11\}$, with the grid search determining $k = 3$ as the optimal choice. Similar ranges and search procedures were applied to the hyperparameters of other classifiers.

The optimal hyperparameters selected for each predictive model are presented in Table 4. These values represent the configurations that resulted in the highest model performance within the specified search space.

**Table 2. 95% confidence interval of the correlation coefficient among the climatic variables with their corresponding p-values.**

| Climatic Variables | Rainfall | Maximum temperature | Minimum temperature | Relative humidity |
|---|---|---|---|---|
| **Rainfall** | $(1, 1)$ | $(-0.58, -0.54)$ | $(0.46, 0.50)$ | $(0.68, 0.71)$ |
| | $< 2.2e-16$ | $< 2.2e-16$ | $< 2.2e-16$ | $< 2.2e-16$ |
| **Maximum temperature** | $(-0.58, -0.54)$ | $(1, 1)$ | $(-0.15-0.09)$ | $(-0.68-0.65)$ |
| | $< 2.2e-16$ | $< 2.2e-16$ | $2.756e-16$ | $< 2.2e-16$ |
| **Minimum temperature** | $(0.46, 0.50)$ | $(-0.15-0.09)$ | $(1, 1)$ | $(0.58, 0.62)$ |
| | $< 2.2e-16$ | $2.756e-16$ | $< 2.2e-16$ | $< 2.2e-16$ |
| **Relative humidity** | $(0.68, 0.71)$ | $(-0.68-0.65)$ | $(0.58, 0.62)$ | $(1, 1)$ |
| | $< 2.2e-16$ | $< 2.2e-16$ | $< 2.2e-16$ | $< 2.2e-16$ |

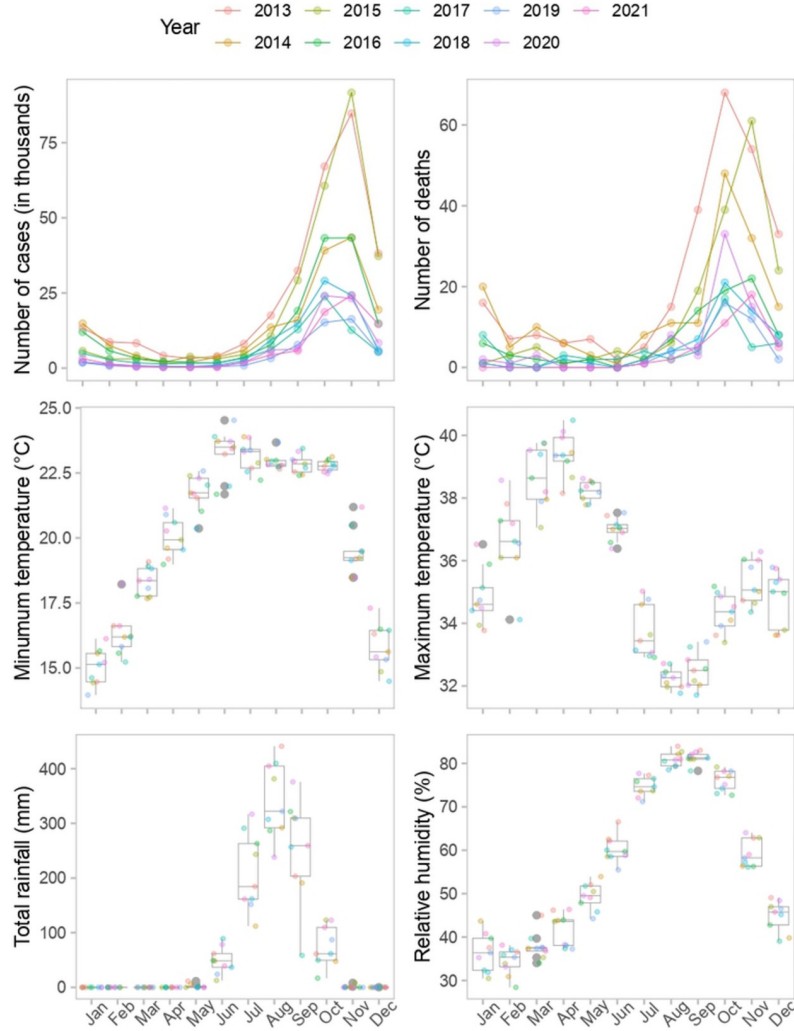

**Fig 5. Variation of the climatic variables, total malaria cases and deaths in each month over the years.** In the upper panels of the first row, we can observe the number of malaria cases and deaths in thousands, although not used as predictors. Each line represents a different year, indicated by the various colors. The bottom four panels show the climate variables and their variability. The boxplots depict each month of the year, while the colored dots represent the data points in the corresponding years. The minimum and maximum temperatures were measured in degrees Celsius (°C), the rainfall was measured in millimeters (mm), and the relative humidity was measured in percentages (%).

**Table 3. Features included in the final predicting models.** The tick ($\checkmark$) mark represents the inclusion of a particular variable in the model, whereas if the variable is not included, the space remains blank. *M* represents months of the calendar year, and *Dis* represents districts and *P* is for population size. $T_{min}$, $T_{max}$, $R$, $H$ are minimum and maximum temperatures, total rainfall, and relative humidity, respectively.

| Models | Features | | | | | | |
|--------|:---:|:---:|:---:|:---:|:---:|:---:|:---:|
| | *M* | *Dis* | $T_{min}$ | $T_{max}$ | *R* | *H* | *P* |
| LR | $\checkmark$ | $\checkmark$ | | | $\checkmark$ | $\checkmark$ | $\checkmark$ |
| KNN | $\checkmark$ | $\checkmark$ | | | | | $\checkmark$ |
| C5.0 DT | $\checkmark$ | $\checkmark$ | $\checkmark$ | $\checkmark$ | | $\checkmark$ | $\checkmark$ |
| ANN | $\checkmark$ | $\checkmark$ | $\checkmark$ | $\checkmark$ | $\checkmark$ | $\checkmark$ | $\checkmark$ |
| RF | $\checkmark$ | $\checkmark$ | | | | | $\checkmark$ |
| XGBoost | $\checkmark$ | $\checkmark$ | $\checkmark$ | $\checkmark$ | $\checkmark$ | $\checkmark$ | $\checkmark$ |
| svmLinear | $\checkmark$ | $\checkmark$ | $\checkmark$ | $\checkmark$ | $\checkmark$ | | $\checkmark$ |
| svmRadial | $\checkmark$ | $\checkmark$ | | | | | $\checkmark$ |

The performance measures across various models on both the training and testing datasets as shown in Tables 5 and 6 respectively reveal insightful patterns. In the training set, Random forest (RF) displayed a strong overall predictive capability (accuracy: 99.8%, sensitivity: 100%, specificity: 98.3%). Artificial Neural Network (ANN) and svmRadial also demonstrated high accuracy, sensitivity and specificty, providing a robust performance on the training set.

C5.0 Decision Trees (C5.0 DT) exhibited exceptional accuracy and specificity (99.9% and 99.3%, respectively) in the training set. On the testing set, C5.0 DT maintained consistent performance (accuracy: 93.3%, sensitivity: 96.7%, specificity: 71.6%), showcasing its effectiveness in practical applications.

Extreme Gradient Boosting (XGBoost) demonstrated remarkable consistency, achieving 100% across all metrics on the training set and maintaining high values on the testing set (accuracy: 93.3%, sensitivity: 96.7%, specificity: 71.6%). In summary, C5.0 DT and XGBoost emerge as the top-performing models, offering a well-balanced and robust predictive capacity for malaria outbreak prediction.

**Table 4. Optimal parameters selected for each classifier.**

| Models | Optimal tuning parameters |
|--------|---------------------------|
| LR | none |
| KNN | k = 3 |
| C5.0 DT | trials = 20 |
| | model = tree |
| | winnow = false |
| ANN | decay = 0 |
| | size = 7 |
| RF | mtry = 46.88889 |
| XGBoost | nrounds = 200 |
| | lambda = 0 |
| | alpha = 0 |
| | eta = 0.3 |
| svmLinear | C = 2.154 |
| svmRadial | C = 1000 |
| | sigma = 100 |

**Table 5. Performance measures on the training set.**

| Models | Accuracy (%) | Sensitivity (%) | Specificity (%) |
|---|---|---|---|
| LR | 88.0 | 98.9 | 52.2 |
| KNN | 93.6 | 100 | 66.9 |
| C5.0 DT | 99.9 | 100 | 99.3 |
| ANN | 96.3 | 99.9 | 78.4 |
| RF | 99.8 | 100 | 98.3 |
| XGBoost | 100 | 100 | 100 |
| svmLinear | 85.9 | 98.6 | 47.8 |
| svmRadial | 97.9 | 100 | 86.2 |

**Table 6. Performance measures on the testing set.**

| Models | Accuracy (%) | Sensitivity (%) | Specificity (%) |
|---|---|---|---|
| LR | 86.8 | 98.8 | 48.6 |
| KNN | 89.4 | 97.7 | 54.9 |
| C5.0 DT | 93.3 | 96.7 | 71.6 |
| ANN | 89.6 | 96.0 | 56.3 |
| RF | 91.5 | 95.4 | 65.1 |
| XGBoost | 93.3 | 96.7 | 71.6 |
| svmLinear | 84.8 | 98.4 | 44.7 |
| svmRadial | 91.0 | 95.0 | 63.7 |

The area under the receiver operating characteristic (ROC) curve (AUC) was also used to compare each model's performance, as shown in Fig 6. The AUC values of all the models are at least 0.9 except for KNN with 0.89 which is the lowest. XGBoost has the highest AUC value of 0.97 and closely followed by C5.0 DT with an AUC of 0.96.

After considering the performance of all the models with respect to the three popular performance metrics (accuracy, sensitivity and specificity) and the AUC values, we recommend the XGBoost and C5.0 DT model as the best performing models. They have the highest accuracy, specificity and AUC values.

## Discussion

Predictive models for malaria outbreaks must be extremely accurate if they are to have any hope of being helpful in clinical and public health decision-making as well as in planning resources for better and more effective services, especially in areas where medical facilities are scarce. With the use of these models, medical professionals might more effectively plan and deliver treatment and diagnostics to lessen the effects of disease on humans, particularly in endemic areas. In this study, we developed machine learning models to estimate the impact of three climatic variables and some non-climatic variables in predicting malaria outbreak in each district in the Gambia.

Two distinct datasets from 2013 to 2021 encompassing climatic and non-climatic factors were analyzed in order to validate and compare all nine models. Fig 4 displays the Pearson correlation among the climatic variables. Tables 5 and 6 displays the sensitivity, specificity, and accuracy of each classifier's performance. Despite the fact that each machine learning technique has unique strengths and weaknesses depending on the application type and size of the

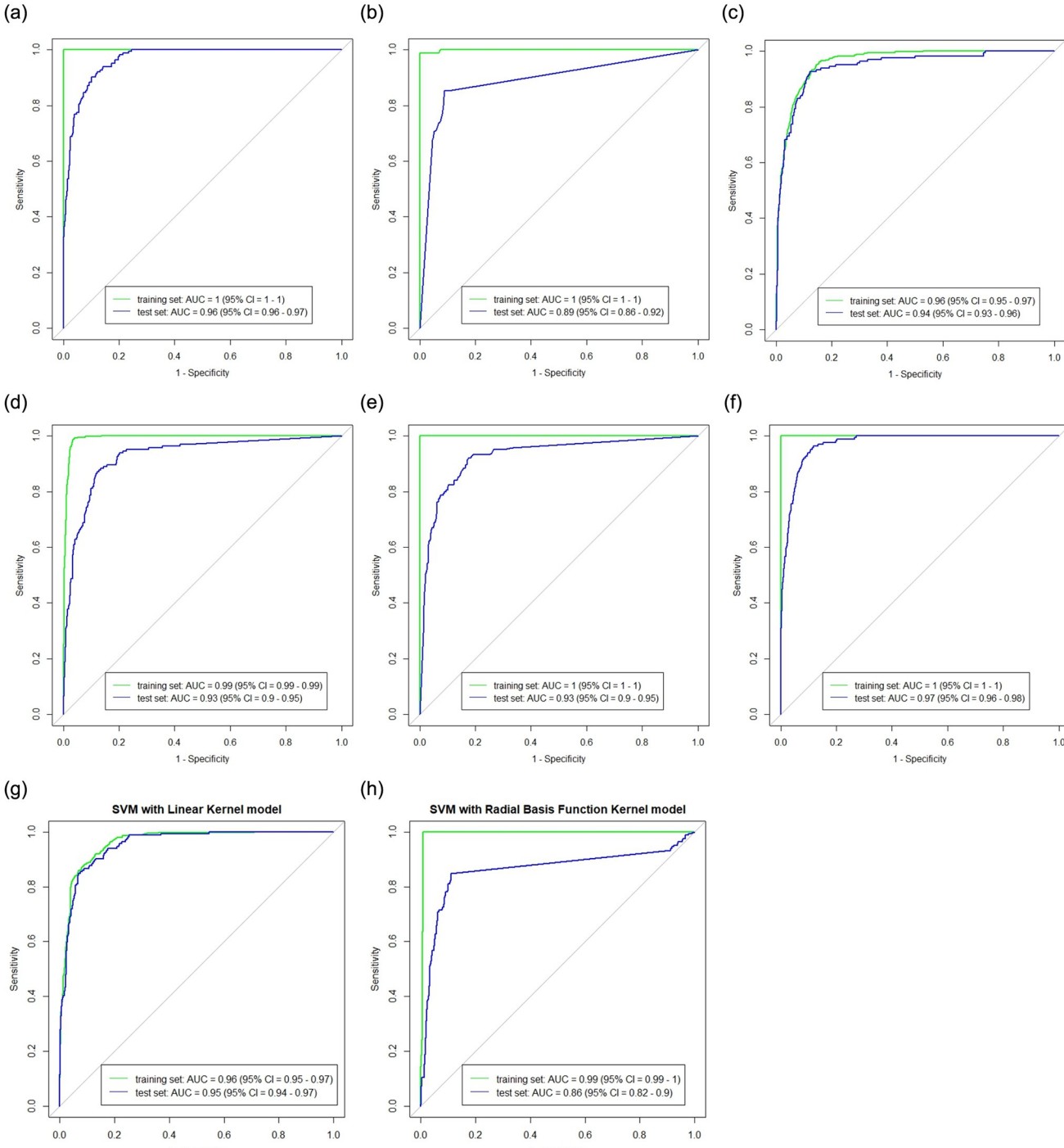

**Fig 6. This plot shows the area under the Receiver Operating Characteristic (ROC), (AUC) of all the learning algorithms.** The green and the blue curve shows the ROC curve using the training and testing data respectively. XGBoost has the highest AUC value of 0.97. **(a)** C5.0 DT. **(b)** KNN. **(c)** Logistic Regression. **(d)** ANN. **(e)** Random Forest. **(f)** XGBoost. **(g)** svmLinear. **(h)** svmRadial.

dataset, all of the models produced promising results in terms of accuracy, sensitivity, specificity as well as AUC values as shown in the result section. We selected XGBoost and C5.0 DT as the best performing models.

We also applied feature selection algorithm technique to return the overall best model with just the relevant features. The non-climatic features (District, population size and month) were forced into all the nine models while the climatic variables were selected by the feature selection algorithm. The variable month was included in all of the predictive models in order to forecast malaria cases for every month in The Gambia. This is because malaria transmission in The Gambia highly varies with respect to the month. It was reported in [23] that The Gambia records its highest clinical cases and fatalities in September and November. This report supports our findings, presented in Fig 5, showing November recording the highest number of malaria cases in The Gambia. The variable District was also fixed to enable us predict malaria cases for each district in The Gambia. From the feature selection results, all the models except for KNN, RF and svmRadial have included at least two climatic features showing why they can be a crucial variables in predicting malaria outbreaks. Each of the climatic variable appeared in four models. The minimum and maximum temperature happened to be selected by the same models. One of the two best models, XGBoost selected all the climatic variables while the second one, C5.0 DT selected all except rainfall. This has further strengthened the significance of these climatic variable in predicting outbreak. Furthermore, these three climatic factors have been linked to the spread of malaria and the development of the anopheles mosquito, according to numerous research [5, 12, 13]. To contextualize our findings, we compare our results with existing literature, shedding light on the effectiveness of machine learning techniques in diverse geographical settings.

Our approach aligns with studies worldwide, such as Kalipe et al. [14], which successfully employed various machine learning techniques in predicting malaria outbreaks in Visakhapatnam, India. Similarly, Zinszer et al. [15] and Lee et al. [16] emphasized the significance of combining environmental and clinical indicators, demonstrating enhanced accuracy in malaria forecasts. These studies underscore the importance of considering diverse variables for robust predictive models, supporting our integration of climatic and non-climatic factors.

Comparing specific models, our results align with Adamu et al. [18], where Random Forest (RF) emerged as the best model. However, it's noteworthy that the optimal model may vary depending on the dataset and geographic location. Our study adds nuance to this understanding, as both XGBoost and C5.0 Decision Trees (C5.0 DT) consistently outperformed other models, providing a well-balanced and robust predictive capacity for malaria outbreak prediction in the Gambia.

It's essential to acknowledge limitations, such as data imbalance and regional variations. These challenges are consistent with broader issues highlighted in the literature. Addressing these limitations requires a nuanced and context-specific approach, emphasizing the need for ongoing research to refine and improve predictive models. Our study, as the first of its kind in the Gambia, lays the groundwork for future endeavors in this critical area, contributing to the global effort to combat malaria and enhance public health outcomes.

## Conclusion and future work

This study examined the significance of machine learning methods for predicting malaria outbreaks in The Gambia. It is crucial to anticipate malaria outbreaks in order to lower the disease's morbidity and fatality rates in The Gambia. And more importantly, the accurate forecasting of malaria outbreak can give public, health authorities and clinical health services

the knowledge they need and the required steps to conduct targeted malaria control strategies that make efficient use of scarce resources to reduce loss of lives.

We employed a number of machine learning techniques to forecast malaria outbreaks in each district in The Gambia. Using historical meteorological data, population size, and malaria case data spanning 9 years (2013–2021), all models were trained, validated, and evaluated. The findings indicated that a combination ofmeteorological and non-climatic factors like population size, month are significant in predicting when malaria epidemics would occur in a particular region or district in The Gambia. Moreover, we recommended XGBoost and C5.0 DT as the best models for prediction based on reasons highlighted in the result section.

In conclusion, our study represents a pioneering effort in malaria prediction within the Gambia. By integrating climatic and non-climatic variables and leveraging effective machine learning models, our work significantly contributes to the growing body of knowledge in malaria forecasting. The insights gained from our research hold particular value for resource-constrained regions, where accurate predictions are essential for informed public health decisions and resource allocation.

This research can be enhanced in the future through the implementation of a hybridized ensemble approach in our machine learning models. By integrating various methodologies, the resulting model is poised to achieve increased reliability and robustness. This step aligns with the evolving landscape of machine learning techniques and ensures our predictive models remain at the forefront of predictive modelling.

Moreover, expanding the scope of our analysis to include additional factors such as treated mosquito nets, indoor residual spray coverage, and other pertinent predictors could contribute significantly to the comprehensive understanding of malaria dynamics in The Gambia. These elements, though not addressed in the current study, represent crucial components that warrant consideration in future investigations.

In essence, this study lays the groundwork for subsequent studies to build upon, incorporating advanced methodologies and expanding the array of variables considered.

## Supporting information

**S1 Fig.**
(TIFF)

## Acknowledgments

The authors would like to acknowledge the support provided by Deanship of Research Oversight and Coordination at King Fahd University of Petroleum & Minerals. We also thank the Health Management Information System (HMIS) office under the ministry of health in The Gambia for providing us with the clinical data. We also thank the Department of Water Resources, the meteorological division under the Ministry of Fisheries, Water Resources and National Assembly Matters in The Gambia for providing us with the climatic data. O. Khan expresses gratitude to Fatoumata Jallow and Omar Ceesay for their tireless efforts in assisting with the dataset acquisition.

## Author Contributions

**Conceptualization:** Ousman Khan, Jimoh Olawale Ajadi.

**Data curation:** Ousman Khan.

**Formal analysis:** Ousman Khan, Jimoh Olawale Ajadi, M. Pear Hossain.

**Funding acquisition:** Jimoh Olawale Ajadi.

**Investigation:** Jimoh Olawale Ajadi, M. Pear Hossain.

**Methodology:** Ousman Khan, Jimoh Olawale Ajadi.

**Software:** Jimoh Olawale Ajadi.

**Supervision:** Jimoh Olawale Ajadi.

**Validation:** M. Pear Hossain.

**Visualization:** M. Pear Hossain.

**Writing – original draft:** Ousman Khan.

**Writing – review & editing:** Jimoh Olawale Ajadi, M. Pear Hossain.

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
