## [Decision Letter · Decision Letter 0]

30 Oct 2023

PONE-D-23-25703Predicting malaria outbreak in The Gambia using machine learning techniquesPLOS ONE

Dear Dr. AJADI,

Thank you for submitting your manuscript to PLOS ONE. After careful consideration, we feel that it has merit but does not fully meet PLOS ONE’s publication criteria as it currently stands. Therefore, we invite you to submit a revised version of the manuscript that addresses the points raised during the review process.

We look forward to receiving your revised manuscript.

Kind regards,

Sathishkumar Veerappampalayam Easwaramoorthy

Academic Editor

PLOS ONE

Journal Requirements:

"This work was supported by the Deanship of Research Oversight and Coordination at King Fahd University of Petroleum and Minerals."

5. We note that [Figure 1] in your submission contain [map/satellite] images which may be copyrighted. All PLOS content is published under the Creative Commons Attribution License (CC BY 4.0), which means that the manuscript, images, and Supporting Information files will be freely available online, and any third party is permitted to access, download, copy, distribute, and use these materials in any way, even commercially, with proper attribution. For these reasons, we cannot publish previously copyrighted maps or satellite images created using proprietary data, such as Google software (Google Maps, Street View, and Earth). For more information, see our copyright guidelines: http://journals.plos.org/plosone/s/licenses-and-copyright.

Reviewers' comments:

Reviewer's Responses to Questions

**Comments to the Author**

1. Is the manuscript technically sound, and do the data support the conclusions?

Reviewer #1: Partly

Reviewer #2: Partly

Reviewer #3: Yes

2. Has the statistical analysis been performed appropriately and rigorously? 

Reviewer #1: Yes

Reviewer #2: Yes

Reviewer #3: Yes

3. Have the authors made all data underlying the findings in their manuscript fully available?

Reviewer #1: No

Reviewer #2: No

Reviewer #3: Yes

4. Is the manuscript presented in an intelligible fashion and written in standard English?

Reviewer #1: Yes

Reviewer #2: No

Reviewer #3: No

5. Review Comments to the Author

Reviewer #1: This paper presents a study on comparing eight machine learning algorithms to predict malaria outbreaks in each district in The Gambia using historical meteorological data.

The model inputs and outputs should be clearly stated.

The 4 climatic variables are shown, but what about the other variables? Are the climatic variables monthly averages or yearly averages? Some clarifications should be given.

Reviewer #2: Although the relevance of the topic, in its current form, the article would need too many adjustments for a major revision. Below are some examples of issues.

The abstract is confusing. For instance, the authors state, "An early warning system that can accurately forecast malaria outbreaks years in advance would be helpful to policymakers to put in measures in reducing morbidity and mortality rate." However, this statement is a justification or conclusion rather than contextualization. This type of argument could be used in the introduction as a justification.

Additionally, the authors present some future research directions in the abstract. I recommend the authors to discuss them in the conclusion.

I recommend avoiding short sentences like "It also performs the worst in specificity analysis."

In the introduction, if possible, could you provide specific statistics on malaria in the Gambia?

The authors could clarify the statement on machine learning and large datasets in the introduction. In the literature, especially for healthcare, there is evidence of the relevant of using machine learning and small datasets,

The authors should state the specific machine learning algorithms they experimented with in the introduction. Additionally, they should justify the choice of such algorithms.

In the introduction, the authors should discuss the challenges of forecasting malaria outbreaks using machine learning.

In Related Works, the authors state, "by combining and aggregating the two datasets." Which are the two datasets?

In Related Works, the authors state, "Extreme gradient boosting (XGBoost) performed better than the other model with 96.26% accuracy." What models?

There is a sentence with no end in the related works section: "Similarly, using clinical data from PubMed abstracts from 1956 to 2019,"

The authors should review the acronym definition. For instance, some acronyms are defined in the incorrect part of the text (in the first mention).

The related works section should include a comprehensive discussion on limitations of previous studies. This discussion would clarify the contribution of the new study.

The following sentence is confusing: "proposed a hybrid classification and regression model to predict the disease outbreak using data on the malaria outbreak,"

Fig. 1 is not cited and explained in the text.

As "several studies [8,22] have shown malaria incidence is influenced by climatic factors such as rainfall, temperature, and humidity," what is the specific contribution of your study?

Did the authors need ethics committee approval to handle the data?

The article must include a more convincing justification for choosing the ML algorithms.

The authors state, "All the models were built using 70% of the data as the training set and the 190 remaining 30% as the testing set." Did you apply holdout or k-fold cross-validation (or both)?

The article needs to include more justification for the choice of performing oversampling.

The article needs to include more justification for the choice of performance metrics.

It needs to be clarified if the authors applied a method for hyperparameter tuning.

The authors should present a more detailed discussion of the specificity results. Are they not relevant to your scenario?

The quality of all figures is poor.

The grammar and spelling need substantial revision. Additionally, clarity and readability need significant improvement. For instance:

However, [9] also reported  However, Kalipe et al. [9] also reported

Please, for all references with the format, for instance "[13] predicts malaria", change for Author [13] or Author et al. [13].

this paper aim  this paper aims

maturity, therefore  maturity.. Therefore,

Support vector machine (SVM)  support vector machine (SVM)

multilayered perceptron  multilayer perceptron

support vector  SVM?

Reviewer #3: Cite current related works.

Justification for data normalization is needed.

The mathematics of the ML models are needed.

Explain upsampling in the context of training and test data (cross-validation).

Why was 10-fold validation used?

Explain the concept of classification versus regression.

Discuss and link with related works.

How can the dataset be validated.

The features of the data should be shown (summary statistics).

6. PLOS authors have the option to publish the peer review history of their article (what does this mean?). If published, this will include your full peer review and any attached files.

Reviewer #1: No

Reviewer #2: No

Reviewer #3: No

---

## [Author Response · Author response to Decision Letter 0]

18 Jan 2024

Reply to comments by Academic Editor on “Predicting malaria outbreak in The Gambia using machine learning techniques” (PONE-D-23-25703).

Thank you for your message. We appreciate the guidance regarding file naming requirements for PLOS ONE. We have thoroughly reviewed our manuscript and taken the necessary steps to ensure that it complies with all the specified requirements, including those related to file naming. 

2. Please note that PLOS ONE has specific guidelines on code sharing for submissions in which author-generated code underpins the findings in the manuscript. In these cases, all author-generated code must be made available without restrictions upon publication of the work. 

We appreciate your diligence in ensuring transparency and reproducibility in the publication process. In our study, we utilized built-in functions from the caret library in R for model training and analysis. These functions are inherent to the R environment and were not custom-authored by us. As such, we don't have specific code snippets to provide for these operations. We have added this in the method section and can be found on page 4.

“In conducting the analysis for prediction, we utilized a robust set of tools and software to ensure accuracy and reliability. All the statistical procedures conducted in this study were implemented using the most recent version of R, a widely recognized open-source statistical tool known for its versatility and extensive data analysis capabilities. Specifically, the predictive modeling and analysis were performed using the caret package \\cite{kuhn2008building} in R. The package provides a comprehensive framework for building, training, and evaluating predictive models, making it particularly suitable for our study's objectives.”

"This work was supported by the Deanship of Research Oversight and Coordination at King Fahd University of Petroleum and Minerals."

This work was supported by the Deanship of Research Oversight and Coordination at King Fahd University of Petroleum and Minerals. The funders had no role in study design, data collection and analysis, decision to publish, or preparation of the manuscript. 

4. In your Data Availability statement, you have not specified where the minimal data set underlying the results described in your manuscript can be found. PLOS defines a study's minimal data set as the underlying data used to reach the conclusions drawn in the manuscript and any additional data required to replicate the reported study findings in their entirety. All PLOS journals require that the minimal data set be made fully available.

The clinical dataset utilized in this study, containing monthly records of malaria cases, deaths, and population size for each district from January 2013 to December 2021, was obtained from the health management information system (HMIS), Directorate of Planning and Information under the Ministry of Health in The Gambia. Access to this dataset is subject to legal and ethical restrictions, and as such, it cannot be made publicly available. However, interested parties may request access to the data by contacting the Directorate of Planning and Information, Ministry of Health, where the data was originally sourced. The program’s manager can be reached on this email: Abdoulie52000@yahoo.com

The meteorological data, including temperature, rainfall, and relative humidity, were collected from the Department of Water Resources, Meteorological Division under the Ministry of Fisheries, Water Resources, and National Assembly Matters in The Gambia. Monthly average readings from nine weather stations across the country were used in the analysis. Unfortunately, due to legal and ethical considerations, we are unable to share the raw meteorological dataset. However, interested researchers can obtain access to this data by contacting the Department of Water Resources, Meteorological Division. The office can be contacted using this email: info@mofwr.gov.gm

We acknowledge the importance of transparency in research, and while we cannot provide unrestricted public access to the datasets, we are committed to facilitating access within the bounds of legal and ethical constraints.

5. We note that [Figure 1] in your submission contain [map/satellite] images which may be copyrighted. All PLOS content is published under the Creative Commons Attribution License (CC BY 4.0), which means that the manuscript, images, and Supporting Information files will be freely available online, and any third party is permitted to access, download, copy, distribute, and use these materials in any way, even commercially, with proper attribution. For these reasons, we cannot publish previously copyrighted maps or satellite images created using proprietary data, such as Google software (Google Maps, Street View, and Earth). For more information, see our copyright guidelines: http://journals.plos.org/plosone/s/licenses-and-copyright.

Thank you for bringing this to our attention. We would like to clarify that the map of the Gambia showing major meteorological stations, presented in Figure 1, was generated using R software as an original creation for the purpose of illustrating the geographical distribution of meteorological stations. No copyrighted material or proprietary data sources were used in the creation of this figure.

We fully understand and respect PLOS' copyright policies and would appreciate further clarification on the specific aspect of the figure that has raised concerns. We are committed to compliance and are open to making any necessary modifications to ensure adherence to copyright guidelines.

Reply to comments by reviewers on “Predicting malaria outbreak in The Gambia using machine learning techniques” (PONE-D-23-25703).

The authors are thankful to the Editor and the anonymous reviewers for providing an opportunity to further improve our paper. The paper is revised by addressing all the suggested points. Here are point by point replies to all comments of the referees (replies are written in bold and the changes in the manuscript are highlighted with yellow). 

Reply to reviewer 1

Reviewer #1: This paper presents a study on comparing eight machine learning algorithms to predict malaria outbreaks in each district in The Gambia using historical meteorological data.

The model inputs and outputs should be clearly stated.

Response: Thank you for your suggestion, the model inputs and outputs have been explicitly outlined in the 'Summary of the Dataset' section under the 'Results,' available in Table 1.

The 4 climatic variables are shown, but what about the other variables? Are the climatic variables monthly averages or yearly averages? Some clarifications should be given.

Response: Thank you for the suggestion. Based on your feedback, we have incorporated a dedicated subsection titled "Climatic Variables and Malaria Incidence Patterns" in the revised version. This new subsection is designed specifically to discuss and present results related to climatic variables, emphasizing their influence and impacts on predicting malaria outbreak in The Gambia. The choice to highlight climatic variables aligns with the study's overarching goal of addressing the intricate challenges in forecasting malaria outbreaks, where the complex interplay of factors, including climatic conditions, necessitates a targeted and nuanced exploration to enhance predictive modelling accuracy. Also, the climate variables are monthly averages and have been adjusted on page 5. 

Reply to reviewer 2

Reviewer #2: Although the relevance of the topic, in its current form, the article would need too many adjustments for a major revision. Below are some examples of issues.

The abstract is confusing. For instance, the authors state, "An early warning system that can accurately forecast malaria outbreaks years in advance would be helpful to policymakers to put in measures in reducing morbidity and mortality rate." However, this statement is a justification or conclusion rather than contextualization. This type of argument could be used in the introduction as a justification.

Response: Thank you for your suggestion. The statement has been removed from the abstract and added to the Introduction.

Additionally, the authors present some future research directions in the abstract. I recommend the authors to discuss them in the conclusion.

Response: Thank you. We have added it in the conclusion. 

“This research can be enhanced in the future through the implementation of a hybridized ensemble approach in our machine learning models. By integrating various methodologies, the resulting model is poised to achieve increased reliability and robustness. This step aligns with the evolving landscape of machine learning techniques and ensures our predictive models remain at the forefront of predictive modelling.

Moreover, expanding the scope of our analysis to include additional factors such as treated mosquito nets, indoor residual spray coverage, and other pertinent predictors could contribute significantly to the comprehensive understanding of malaria dynamics in The Gambia. These elements, though not addressed in the current study, represent crucial components that warrant consideration in future investigations.

In essence, this study lays the groundwork for subsequent studies to build upon, incorporating advanced methodologies and expanding the array of variables considered.”

I recommend avoiding short sentences like "It also performs the worst in specificity analysis."

Response: We have made the necessary correction.

In the introduction, if possible, could you provide specific statistics on malaria in the Gambia?

Response: Yes, it is possible. We have added some statistics on malaria in The Gambia in the Introduction. Please find it in line no. 10-20.

The authors could clarify the statement on machine learning and large datasets in the introduction. In the literature, especially for healthcare, there is evidence of the relevant of using machine learning and small datasets.

Response: Thank you for the suggestion. We have revised the introduction to provide clarification on machine learning's relevance to both large and small datasets in the healthcare literature. Please find the changes in line no. 25-30.

The authors should state the specific machine learning algorithms they experimented with in the introduction. Additionally, they should justify the choice of such algorithms.

Response: We have addressed your suggestion by explicitly stating the machine learning algorithms used in the introduction. Furthermore, we have provided justification for the selection of these algorithms. All changes could be found in line no. 41-46.

In the introduction, the authors should discuss the challenges of forecasting malaria outbreaks using machine learning.

Response: We have added the challenges of forecasting malaria outbreaks using machine learning in the Introduction.

In Related Works, the authors state, "by combining and aggregating the two datasets." Which are the two datasets?

Response: We appreciate your observation. In the updated manuscript, we have explicitly mentioned that the two datasets refer to historical meteorological data and records of malaria cases. Please find these changes in the Related work section and line number 60-63. 

In Related Works, the authors state, "Extreme gradient boosting (XGBoost) performed better than the other model with 96.26% accuracy." What models?

Response: Thank you for the observation. The XGBoost model was compared with the various machine learning models such as K-nearest neighbor, Naïve Bayes, support vector machine, …etc.. We observed XGBoost performed better than the other models with 96% accuracy. 

There is a sentence with no end in the related works section: "Similarly, using clinical data from PubMed abstracts from 1956 to 2019,"

Response: Thank you for the observation. It has been updated.

The authors should review the acronym definition. For instance, some acronyms are defined in the incorrect part of the text (in the first mention).

Response: Thank you for the suggestion. We have carefully reviewed and adjusted all the acronym definitions.

The related works section should include a comprehensive discussion on limitations of previous studies. This discussion would clarify the contribution of the new study.

Response: Thank you for the insight. We have updated the related work section as per your suggestion and can be found in line no. 118-125.

The following sentence is confusing: "proposed a hybrid classification and regression model to predict the disease outbreak using data on the malaria outbreak,"

Response: Thank you for noticing. We have clarified it.

Fig. 1 is not cited and explained in the text.

Response: 

Thank you for bringing this to our attention. We would like to clarify that the map of the Gambia showing major meteorological stations, presented in Figure 1, was generated using R software as an original creation for the purpose of illustrating the geographical distribution of meteorological stations. No copyrighted material or proprietary data sources were used in the creation of this figure. We have cited the package that we employed.

As "several studies [8,22] have shown malaria incidence is influenced by climatic factors such as rainfall, temperature, and humidity," what is the specific contribution of your study?

Response: The specific contribution of our study has been added on the “Dataset used” section on page 5.

“The specific contribution of our study lies in utilizing two distinct datasets (historical meteorological and clinical datasets) spanning nine years (January 2013 to December 2021) to predict malaria outbreaks in each district of The Gambia. While previous studies have highlighted the influence of climatic factors such as rainfall, temperature, and humidity on malaria incidence \\cite{thomson2005use,ceesay2010continued}, our approach integrates machine learning techniques to provide a more accurate and district-specific prediction of malaria outbreaks. This extends the existing knowledge by leveraging advanced analytical methods for a targeted and nuanced understanding of the impact of climatic conditions on malaria transmission at a local level.”

Did the authors need ethics committee approval to handle the data?

Response: The dataset utilized in our study was provided by the relevant authorities, specifically the Ministry of Health and the Department of Water Resources. As the data involved anonymized and aggregated information and was obtained through official channels, no formal ethical approval was required for its usage. We ensured strict adherence to data protection regulations and maintained the confidentiality and privacy of the information throughout the study.

The article must include a more convincing justification for choosing the ML algorithms.

Response: Thank you for your suggestion. We have given a general justification for the use of the ML algorithms in the “Prediction models” subsection on page 6.

The authors state, "All the models were built using 70% of the data as the training set and the 190 remaining 30% as the testing set." Did you apply holdout or k-fold cross-validation (or both)?

Response: We utilized a combination of both methods. Initially, the holdout method was employed by partitioning the data into a training set (70%) and a testing set (30%) for initial validation. While training the model exclusively on the 70% training set, we implemented 10-fold cross-validation to mitigate the risk of overfitting.

The article needs to include more justification for the choice of performing oversampling.

Response: We have added the justification. It can be found on page 9. Thank you.

“By implementing up-sampling in this manner, the training data becomes more representative of the underlying distribution of outbreak and no outbreak cases. This contributes to a more balanced learning process during model training, allowing the machine learning algorithm to better capture patterns within the minority class. It's essential to note that up-sampling is a training-specific technique and does not directly influence the distribution of classes in the testing dataset or during cross-validation.

In summary, the use of up-sampling in the training dataset helps mitigate the challenges posed by class imbalance, fostering improved model generalization and predictive accuracy during the training phase while preserving the natural distribution of classes in the testing dataset and during cross-validation.”

The article needs to include more justification for the choice of performance metrics.

Response: We have added the justification and can be found on page 8.

“These selected performance metrics collectively offer a thorough evaluation of the models, capturing aspects of accuracy, sensitivity, specificity, and discriminatory capability. Their inclusion ensures a multifaceted assessment aligned with the diverse requirements of malaria outbreak prediction.”

It needs to be clarified if the authors applied a method for hyperparameter tuning.

Response: We performed hyperparameter tuning using the grid search method and have incorporated it into the script on page 12.

“To optimize the performance of each predictive model, hyperparameter tuning was conducted using the grid search method. This approach involves systematically searching through a predefined set of hyperparameter values to identify the combination that yields the optimal model performance.

For instance, the parameter $k$ in the K-Nearest Neighbors (KNN) classifier was explored across the values $K=\\{3,5,7,9,11\\}$, with the grid search determining $k=3$ as the optimal choice. Similar ranges and search procedures were applied to the hyperparameters of other classifiers.”

The authors should present a more detailed discussion of the specificity results. Are they not relevant to your scenario?

Response: The specificity results are relevant. We have added a detailed explanation of the specificity results on page 12.

The quality of all figures is poor.

The grammar and spelling need substantial revision. Additionally, clarity and readability need significant improvement. For instance:

However, [9] also reported  However, Kalipe et al. [9] also reported

Please, for all references with the format, for instance "[13] predicts malaria", change for Author [13] or Author et al. [13].

this paper aim  this paper aims

maturity, therefore  maturity.. Therefore,

Support vector machine (SVM)  support vector machine (SVM)

multilayered perceptron  multilayer perceptron

support vector  SVM?

Response: Thank you for your suggestion. We have taken care of the issues raised and have improved the grammar also.

Reviewer #3: Cite current related works.

Response: Thank you for your feedback. We have incorporated some additional references to current related works in the revised manuscript. 

“In Senegal, Diao et al. \\cite{diao2023generalized} conducted a significant study in malaria forecasting. They formulated a generalized linear model based on Poisson and negative binomial regression models, considering climatic variables, insecticide-treated bed-nets distribution, Artemisinin-based combination therapy, and historical malaria incidence. The study demonstrated the efficacy of the Poisson regression model and addressed issues of over-forecasting through the saturation of rainfall. In Rajasthan, India, Singh et al. \\cite{singh2023leveraging} proposed a hybrid ML algorithm (P2CA $-$ PSO $-$ ANN) for malaria outbreak prediction in districts like Barmer, Bikaner, and Jodhpur. Using meteorological variables, data fusion, and P2CA, the model achieved accurate predictions, outperforming benchmarks. It shows promise as an early warning system based solely on meteorological data.”

Justification for data normalization is needed.

Response: Thank you for your suggestion. We have added the justification for using data normalization on page 5.

“The dataset underwent a comprehensive normalization process to enhance the effectiveness of machine learning algorithms. It is a critical preprocessing step that plays a pivotal role in enhancing the quality and effectiveness of machine learning algorithms. In this study, normalization was applied to numerical independent variables using the min-max normalization technique. This process transforms features into a shared range, mitigating the impact of larger numeric values dominating the model's learning process. Normalization becomes particularly crucial in scenarios where the scale of numerical features varies significantly. Without normalization, features with larger numeric values may exert undue influence on the learning algorithm, potentially overshadowing the contributions of smaller-scale features. The rationale behind data normalization lies in its ability to create a level playing field for numerical features, ensuring that each contributes proportionally to the learning process. This contributes to the overall robustness and reliability of the machine learning models employed in predicting malaria outbreaks. The normalization formula employed in this study, as depicted in Eq.\\ref{eq:1}, ensures that all numerical variables are scaled to values between 0 and 1.”

The mathematics of the ML models are needed.

Response: Thank you for your feedback. We have added some mathematics of each of the ML models.

Explain upsampling in the context of training and test data (cross-validation).

Response: Thank you for your suggestion. We have added a detailed explanation of upsampling in the context of training and test data (cross-validation) on page 9. 

“In the context of training and testing data, the initial random split of the dataset into these sets highlighted a significant class imbalance, particularly regarding "outbreak" and "no outbreak" cases, as depicted in Fig \\ref{fig:outbreak proportion}. Notably, 87\\% of both the training and testing datasets consisted of "no outbreak" cases.”

Why was 10-fold validation used?

Response: We have added the reason of using 10-fold cross validation on page 9.

“The choice of 10-fold cross-validation was motivated by its effectiveness in providing a robust evaluation of model performance. By dividing the training data into ten folds, our models underwent comprehensive training and testing iterations, promoting thorough exposure to diverse data patterns. This approach mitigates overfitting, enhances the reliability of performance estimates, and ensures a resource-efficient use of the available data. The decision to repeat the 10-fold cross-validation five times further strengthens the statistical significance of our performance evaluations.”

Explain the concept of classification versus regression.

Response: Thank you for your valuable suggestion. We recognize the importance of clarifying the concepts of classification and regression in the context of our paper. In response, we have provided a brief explanation in the “Prediction models” sections on page 9 to enhance clarity and comprehension.

“Chosen for their widespread use and high predictive accuracy, these models are versatile and commonly employed in both classification and regression tasks. In the context of machine learning, classification involves categorizing instances into predefined classes, such as predicting whether a district is likely to experience a malaria outbreak. On the other hand, regression predicts continuous numerical values. Our study primarily focuses on a classification problem.”

Discuss and link with related works.

Response: Thank you for your suggestion. We have updated the discussion as per your suggestion. 

“Our approach aligns with studies worldwide, such as Kalipe et al. \\cite{kalipe2018predicting}, which successfully employed various machine learning techniques in predicting malaria outbreaks in Visakhapatnam, India. Similarly, Zinszer et al. \\cite{zinszer2015forecasting} and Lee et al. \\cite{lee2021machine} emphasized the significance of combining environmental and clinical indicators, demonstrating enhanced accuracy in malaria forecasts. These studies underscore the importance of considering diverse variables for robust predictive models, supporting our integration of climatic and non-climatic factors.

Comparing specific models, our results align with Adamu et al. \\cite{adamu2021malaria}, where Random Forest (RF) emerged as the best model. However, it's noteworthy that the optimal model may vary depending on the dataset and geographic location. Our study adds nuance to this understanding, as both XGBoost and C5.0 Decision Trees (C5.0 DT) consistently outperformed other models, providing a well-balanced and robust predictive capacity for malaria outbreak prediction in the Gambia.”

How can the dataset be validated.

Response: We have added how it can be validated in the “Data cleaning and preprocessing” section on pages 5 and 6.

The features of the data should be shown (summary statistics).

Response: Thank you for your suggestion. We have added a summary statistic of the dataset used in Table 1 on page 10.

---

## [Editor Report · Decision Letter 1]

9 Feb 2024

Predicting malaria outbreak in The Gambia using machine learning techniques

PONE-D-23-25703R1

Dear Dr. AJADI,

We’re pleased to inform you that your manuscript has been judged scientifically suitable for publication and will be formally accepted for publication once it meets all outstanding technical requirements.

Kind regards,

Sathishkumar Veerappampalayam Easwaramoorthy

Academic Editor

PLOS ONE
---

## [Editor Report · Acceptance letter]

1 Mar 2024

PONE-D-23-25703R1 

PLOS ONE

Dear Dr. Ajadi, 

I'm pleased to inform you that your manuscript has been deemed suitable for publication in PLOS ONE. Congratulations! Your manuscript is now being handed over to our production team.

Kind regards, 

on behalf of

Dr. Sathishkumar Veerappampalayam Easwaramoorthy 

Academic Editor

PLOS ONE